# The Importance of Kinases in Retinal Degenerative Diseases

**Paulo F. Santos** [1,2,3,4] ![ID], **António Francisco Ambrósio** [1,2,3,5] ![ID] and **Hélène Léger** [1,2,3,*] ![ID]

1. Coimbra Institute for Clinical and Biomedical Research (iCBR), Faculty of Medicine, University of Coimbra, Azinhaga de Santa Comba, Celas, 3000-548 Coimbra, Portugal; pfsantos@ci.uc.pt (P.F.S.); afambrosio@fmed.uc.pt (A.F.A.)
2. Center for Innovative Biomedicine and Biotechnology (CIBB), Faculty of Medicine, University of Coimbra, Rua Larga, 3004-504 Coimbra, Portugal
3. Clinical Academic Center of Coimbra (CACC), 3000-370 Coimbra, Portugal
4. Department of Life Sciences, University of Coimbra, CC Martim de Freitas, 3000-456 Coimbra, Portugal
5. Association for Innovation and Biomedical Research on Light and Image (AIBILI), Azinhaga de Santa Comba, Celas, 3000-548 Coimbra, Portugal
* Correspondence: hleger@uc.pt; Tel.: +351-239480287

**Abstract:** Kinases play crucial roles in the pathophysiology of retinal degenerative diseases. These diseases, such as diabetic retinopathy, age-related macular degeneration, glaucoma, and retinitis pigmentosa, are characterized by progressive degeneration of retinal cells, including photoreceptors, ganglion cells, vascular cells, and retinal pigment epithelium, among others. The involvement of kinases in cell survival and apoptosis, immune responses and inflammation regulation, mitochondrial functions and mitophagy, autophagy, and proteostasis is crucial for maintaining cellular homeostasis and responding to various stressors. This review highlights the importance of studying kinases to better understand their functions and, regulation permitting, enable the identification of novel molecular players or potential drug targets and, consequently, the development of more effective and precise treatments to slow or halt the progression of retinal degenerative diseases.

**Keywords:** retina; retinal degenerative diseases; kinases; nuclear Dbf2-related (NDR) kinases





## 1. Introduction

All cells need to tailor their metabolism to their environment by carrying extracellular information to their cytoplasm and nucleus in a process called cell signaling. Cell signaling is essential for all critical cellular processes, such as controlled response to stimuli, cell growth and division, cell fate, migration, and cell–cell coordination. Malfunctions in cell signaling are the cause of many diseases, ranging from cancer and immune diseases to neurodegenerative diseases [1].

Cell signaling understanding started with the identification of reversible protein transformations. Phosphorylation is one of these reversible protein posttranslational modifications involved in cell signaling. In 1906, Phoebus Levene identified phosphate residue in the vitellin. Forty years later, Carl and Gerty Cori described two forms of glycogen phosphorylase. The first indications of the phosphorylation/dephosphorylation mechanism were obtained in the 1950s by Edmond Fischer and Edwin Krebs, who were later recognized with the Nobel Prize in Physiology or Medicine in 1992, as well as in the works published at the same time by Walter Wosilait and Earl Sutherland [2]. Phosphorylation is the transfer of a phosphate group from high-energy, phosphate-donating molecules, such as ATP, to specific substrates (proteins, lipids, or carbohydrates) by phosphotransferases called kinases. During the "decade of protein kinase cascades", in the 1990s, many studies demonstrated the importance of the reversible phosphorylation by kinases in the regulation of a molecule's activity, reactivity, and ability to bind other molecules, making this transformation essential for the regulation of many cellular processes, such as metabolism, cell signaling, protein regulation, cellular transport, and secretory processes [3–5].

Kinases are one of the largest groups of enzymes in living organisms. The human kinome—determined as the complete set of kinases encoded in its genome—contains more than 500 different kinases (>1.5% of the human genome), with a wide diversity of structures, substrates, and functions, as well as high complexity. Indeed, a specific kinase can often have multiple substrates, and multiple proteins can serve as the substrate for more than one specific kinase [6].

The visual system is the command center for processing visual information and is composed of the eyes and the visual cortex. Humans, like many other animals, are highly visual beings, as they depend on visual information for basic behaviors, such as finding food, communication, defense, or mate finding, as well as for more complex behaviors, such as parental care and socialization.

Vision is a process that begins in the eye, a highly specialized organ that captures light, and particularly in the retina, which transforms light into electrochemical signals that are then transmitted, via the optic nerve, to the visual cortex, where they are processed. Retinal degenerative diseases, such as diabetic retinopathy, age-related macular degeneration, glaucoma, or retinitis pigmentosa, can lead to irreversible vision loss and are a significant public health concern, affecting more than 400 million people worldwide—including more than 200 million people affected by age related macular degeneration (AMD), a number that will increase significantly in the coming years [7,8]. Kinases, as crucial components of cellular signaling pathways, play an important role in the pathogenesis, progression, and potential treatment of these diseases.

It is said that the eyes are the "window to the brain". Therefore, it is fundamental to better understand the molecular mechanisms underlying the physiology and pathologies of the eye as a vital sensory organ for vision, but also as a model through which we can gain insights into the physiology and pathology of the brain. In this review, we will discuss the importance of kinases in research related to retinal degenerative diseases, providing a concise inventory of the diverse categories of kinases and their involvement in retinal homeostasis and degenerative diseases. We will highlight the importance of identifying and developing new therapeutic drugs and targeted therapies to mitigate vision loss in affected individuals.

## 2. Presentation of the Visual System

The visual system is composed of the eyes connected to the visual cortex by the optic nerve. The eye is a complex organ composed of several components, each with a specific function: the cornea acts as a protective transparent outer lens that refracts light; the iris—the colored part of the eye—regulates the amount of light that enters the eye via the pupil, which can expand or contract; the lens adjusts the focus of light onto the retina, a multicellular layer located at the back of the eye; and the retinal pigment epithelium (RPE) and the choroid are in charge of nourishing and protecting the visual cells. The retina is the neuronal part of the eye where the actual visual processing takes place (Figure 1).

In particular, the retina contains specialized light-sensitive cells—the photoreceptors—which convert incoming light into a cascade of electrical impulses that travel through the retina and the optic nerve to the brain. Photoreceptors are highly energy-consuming specialized neurons containing photosensitive molecules in the outer segment. These molecules, called opsins, transform light into chemical and electrical signals. One of the critical components of phototransduction—the process by which light is converted into electrical signals—is the cyclic guanosine $3',5'$-monophosphate signaling pathway. Cyclic guanosine $3',5'$-monophosphate (cGMP) activates the kinase known as cGMP-dependent protein kinase G (PKG) [9–11]. Various retinal degenerative diseases affecting the photoreceptors present dysregulation of cGMP signaling, leading to an overload of calcium ions ($Ca^{2+}$) and the activation of the PKG, promoting cell death signaling pathways (Figure 2). It is therefore important to understand the mechanisms underlying this process to develop new therapeutic approaches for photoreceptor protection targeting cGMP/PKG signaling in retinal degenerative diseases [9–11].

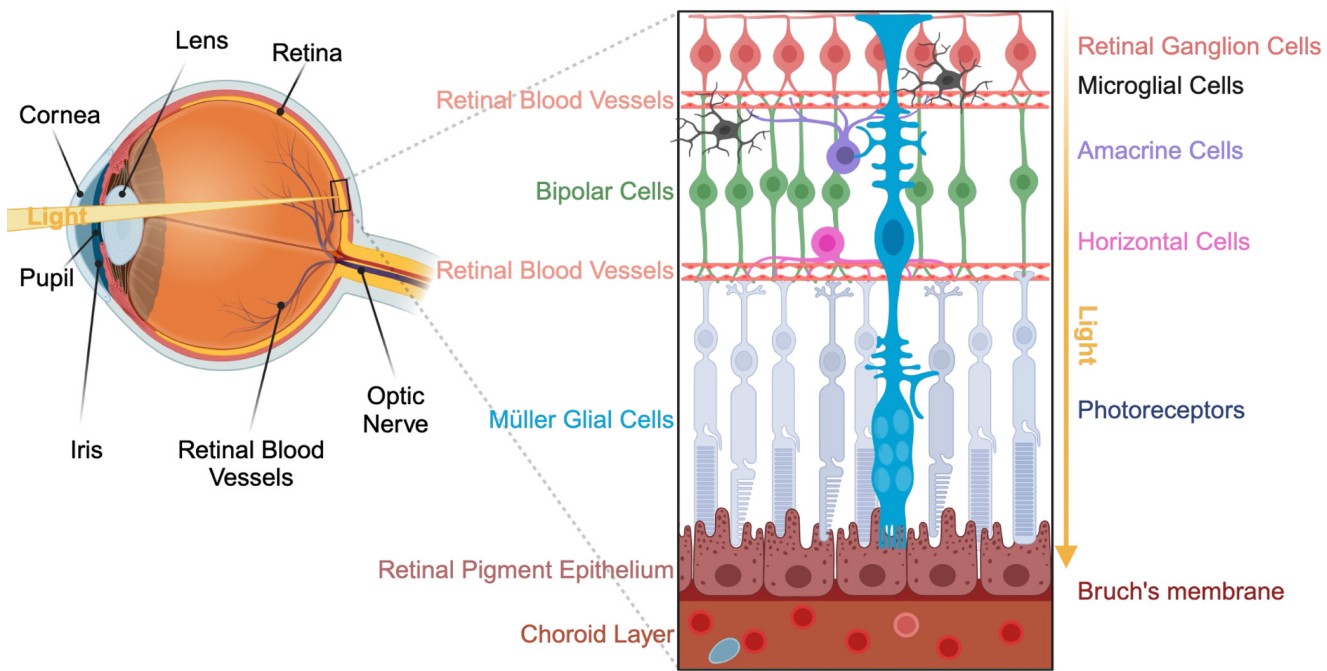

**Figure 1.** Structure of the eye. Schematic of the eye cup (**left**) and retina organization (**right**). Created with BioRender.com.

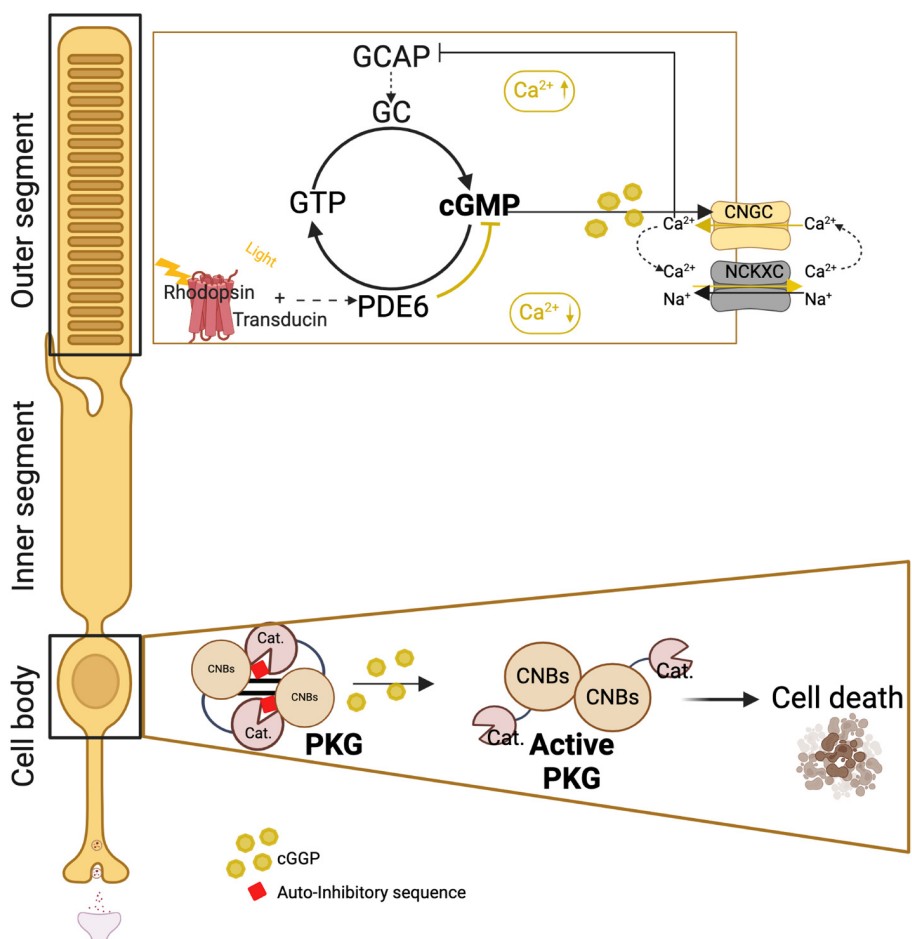

**Figure 2.** Schematic of the regulation of the phototransduction cascade, which depends on the interplay between cyclic guanosine-monophosphate (cGMP) and calcium ions ($Ca^{2+}$). In the darkness,

Ca$^{2+}$ blocks the activation of retinal guanylyl cyclase (GC) by the guanylyl cyclase-activating protein (GCAP). GC produces cGMP, which opens the cyclic nucleotide-gated channel (CNGC), inducing Ca$^{2+}$ influx. In light, phototransduction promotes the activation of cGMP-phosphodiesterase 6β (PDE6B) and the hydrolyses of cGMP, causing the CNGC to close, and the activation of cGMP synthesis. In conclusion, Ca$^{2+}$ influx leads to the inhibition of cGMP via a regulatory feedback loop, keeping the levels of both cGMP and Ca$^{2+}$ in physiological ranges and ensuring proper phototransduction. However, when this regulatory loop is unbalanced, resulting in increased Ca$^{2+}$ influx or Ca$^{2+}$ depletion, depending on the gene of the loop affected by a mutation, high levels of cGMP are found in the outer segment, leading to the overactivation of the cGMP-dependent protein kinase G (PKG) in the cell body and the death of the photoreceptors. However, the precise mechanism of the PKG in photoreceptor cell death is not well characterized, and future studies are necessary to identify which PKG substrates would be a putative target for future therapy development. Adapted from [9–11] and created with BioRender.com.

The visual pigments of the photoreceptors need to be recycled by the retinal pigment epithelium after the phototransduction. Both processes—the phototransduction and the recycling of the visual pigments—demand extensive energy, generating a high level of reactive oxygen species (ROS). With aging, genetic impairments, or an oxidative environment, the repair of the ROS-induced damages is unable to compensate for the production of the ROS, thus causing oxidative stress, leading over time to neurodegeneration and retinal diseases [12]. Numerous studies have shown that oxidative stress can lead to the activation of specific kinase signaling, such as the mitogen-activated protein kinase (MAPK) or the c-Jun N-terminal kinase (JNK) pathways, which can activate either cell survival or apoptosis [13].

Kinases are important enzymes in ophthalmology. Kinase inhibitors could have a therapeutic advantage in treating certain ocular diseases by modulating specific kinase signaling pathways involved in disease progression. It is therefore crucial to better understand the kinome of the visual system to identify new drugs and improve existing treatments for ocular diseases. Kinases are divided into categories based on their type of substrate: lipid kinases, carbohydrate kinases, and protein kinases. Kinases can be further subdivided based on the amino acid they phosphorylate and the existence of isoenzymatic forms (Figure 3).

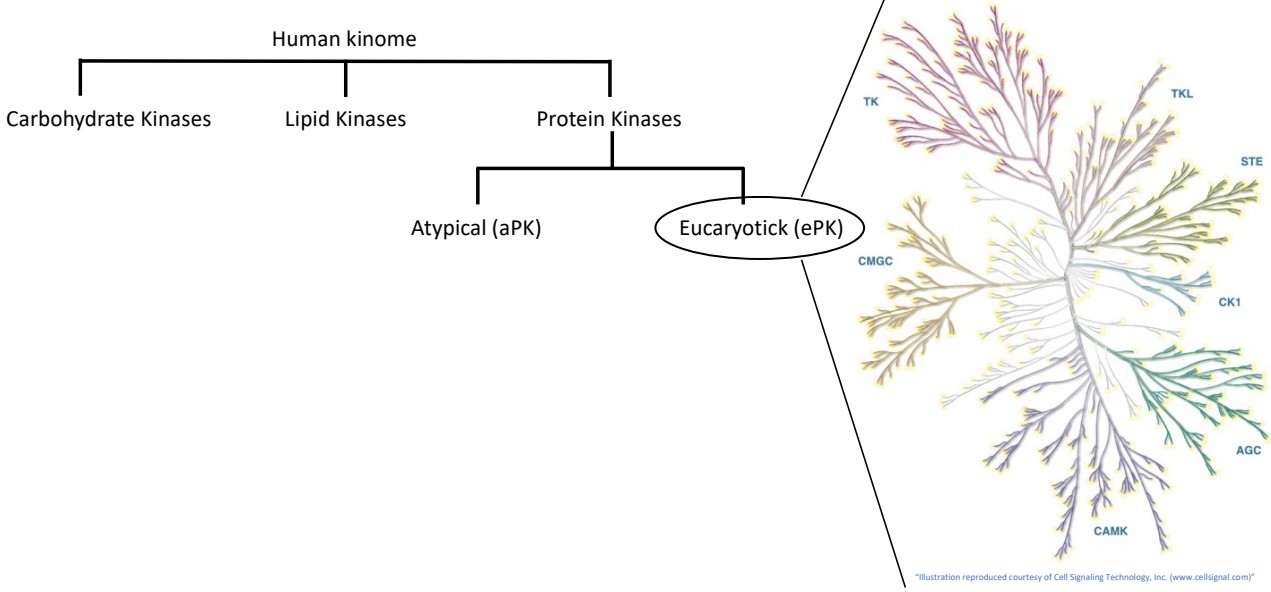

**Figure 3.** The human kinome. Based on http://www.kinhub.org/kinmap/index.html (Accessed on 15 November 2023). The kinome refers to the complete set of kinases of an organism. The human

kinome comprises more than 500 kinases, which are classified based on their type of substrate as carbohydrate kinases, lipid kinases, and protein kinases. Protein kinases can be further subdivided into two main groups: atypical kinases and eukaryotic protein kinases. Atypical kinases represent a diverse group of enzymes that exhibit kinase activity but may not conform to the typical structural features of classical protein kinases or eukaryotic protein kinases, often lacking some of the conserved motifs found in classical protein kinases. Their structures can vary significantly. Eukaryotic protein kinases or classical protein kinases constitute the largest group of protein kinases. The eukaryotic protein kinases (right) share a conserved catalytic domain that notably includes an ATP-binding site and an activation loop. Illustration of the eukaryotic protein kinases tree reproduced by courtesy of Cell Signaling Technology. Created with BioRender.com (Accessed on 15 November 2023).

## 3. The Human Kinome

### 3.1. The Carbohydrate Kinases

Carbohydrate kinases activate monosaccharides by adding a phosphate group. The carbohydrate kinases utilize a common strategy to add a phosphate group to the sugar hydroxyl by deprotonating the reactive hydroxyl, which can then attack the donor phosphate. This modification, a very ancient reaction, is central to the metabolism of carbohydrates. Modern organisms contain carbohydrate kinases from at least five main protein families: (1) hexokinase and glucokinase, (2) pyruvate kinase, (3) AMP-activated protein kinase (AMPK), (4) phosphofructokinase (PFK), and (5) glycogen synthase kinase-3 (GSK-3). Defects in carbohydrate phosphorylation have been associated with several diseases, such as juvenile cataracts in weeks-old babies and in 30–50% of cases of onset diabetes in young patients [14]. Carbohydrates are an important target for novel therapeutics [14]. The carbohydrate kinases represent a very small percentage of the kinome [14].

Amongst the carbohydrate kinases, the hexokinases and glucokinases perform the key step of glucose metabolism by phosphorylating glucose during its entry into the cells [14]. The pyruvate kinases catalyze the final step of glycolysis, converting phosphoenolpyruvate to pyruvate and producing ATP. The adenosine monophosphate-activated protein kinase (AMPK) is a master regulator of cellular energy homeostasis, influencing carbohydrate metabolism by regulating processes like glucose uptake, glycolysis, and gluconeogenesis. AMPK is activated in response to an increase in AMP levels, indicating a cellular energy deficit [9].

The phosphofructokinases are a key regulatory enzyme in glycolysis. They catalyze the conversion of fructose-6-phosphate to fructose-1,6-bisphosphate. Finally, glycogen synthase kinase-3 (GSK-3) is involved in the regulation of glycogen synthesis. It phosphorylates and inhibits glycogen synthase, a key enzyme in glycogen production and glycogen storage [15].

The rod and cone photoreceptor cells have high energy expenditure, almost equivalent to that of a cancer cell. Photoreceptor cell survival is centered around aerobic glycolysis, which converts glucose to lactate transported to RPE and Müller cells. In the RPE and Müller cells, lactate is converted to pyruvate, fueling the mitochondria for oxidative phosphorylation. Ablation of this pathway causes retinal degeneration, such as age-related macular degeneration, while aerobic glycolysis is essential for normal rod function and its upregulation is neuroprotective. It has been demonstrated that the stimulation of AMPK by metformin has a neuroprotective effect on photoreceptors and RPE in mouse models of retinal degeneration [16,17].

Recently, it was demonstrated that the carbohydrate kinase hexokinase-2 (HK2) plays an important role in the development of insulin resistance and vascular complications in diabetes when stabilized by glucose. This induces an increased flux of glucose metabolism without a change in the expression and activity of glycolytic enzymes, causing a PIC of increased glycolytic intermediates that lead to mitochondrial dysfunction and increased reactive oxygen species (ROS) formation, activation of hexosamine and protein kinase C pathways, increased formation of methylglyoxal-producing dicarbonyl stress, and activation of the unfolded protein response. HK2-linked glycolytic overload and unscheduled glycolysis are perfect examples of the importance of kinases in metabolism and diseases.

These reactions and their metabolic consequences are found in the vasculature, kidneys, retina, and peripheral nerves and in early-stage embryo development in diabetes, and they likely sustain the development of diabetic vascular complications and embryopathy. Further studies are required to develop treatment for the vascular complications of diabetes [18].

The retina expresses both PKM1 in the inner plexiform layer and ganglion cell layer and PKM2 in the inner segments and outer plexiform layer of the photoreceptors. The functional role of PKM1 in the retina is currently unknown. Loss of PKM2 causes the accumulation of glycolytic intermediates, decreased pyruvate and lactate levels, and an increase in NADPH levels. Moreover, PKM2 is important in the regulation of photoreceptor-specific proteins, such as cGMP-PDE6B and GTPase-activating protein, both regulators of G-protein signaling 9 (RGS9). PDE6B is a key regulator of the cGMP and the cGMP-gated channel at the plasma membrane of the outer segments, controlling the polarization of the rod photoreceptors and thus the phototransduction kinetics. Taking inspiration from cancer treatment, modulating PKM2 levels could be an interesting approach to promote photoreceptor survival among patients affected by retinal degenerative diseases such as retinitis pigmentosa. This would contribute to preventing retinal cell death by reducing the anabolic activity and fueling mitochondria for oxidative phosphorylation and to promoting retinal cell survival [16].

Research on the PFK in retina is not well developed (18 papers). However, recent publications have demonstrated that the hypoxia-inducible factor $1\alpha$ (HIF1$\alpha$)-6-phosphofructo-2-kinase/fructose-2,6-bisphosphatase 3 (PFKFB3) pathway is central to the pathophysiology of diabetic retinopathy regarding pathologic angiogenesis and neurodegeneration. PFKFB3 is a key player in the regulation of endothelial tip-cell competition, thereby regulating sprouting angiogenesis. PFKFB3 is also highly involved in glycolysis, modulating the antioxidative capacity of neurons and eventually leading to neuronal loss and reactive gliosis [19].

A PubMed search for "glycogen synthase kinase-3 (GSK3)" and "retina" showed less than 100 publications. However, GSK3 is a key regulator in various cell signaling processes in the nervous system, such as (i) apoptosis through the promotion of p53-mediated apoptosis and inhibition of pro-survival transcription factors; (ii) phosphorylating and inhibiting glycogen synthase (GS) and therefore limiting the conversion of glucose into glycogen; (iii) promoting retinal oxidative stress via the negative regulation of the nuclear factor erythroid-2-related factor 2; and (iv) negatively regulating anti-inflammatory cytokine production, such as IL-2, IL-10, IL-22, and IL-33, and positively regulating pro-inflammatory cytokines and chemokines, such as TNF-$\alpha$, interleukin (IL-)1$\beta$, IL-6, IL-17, C-C motif chemokine ligands (CCLs) 2 and 12, and C-X-C motif chemokine ligands (CXCLs) 1 and 10. Finally, inhibition of GSK3 can (v) rescue abnormal retinal vasculature and (vi) promote optic nerve regeneration via the enhanced activity of key downstream effectors, like CRMP2 and mTOR. Therefore, a rising number of studies have highlighted the interest of developing therapeutics targeting GSK3 to treat neurodegenerative diseases [15].

### 3.2. The Lipid Kinases

Lipid kinases are enzymes that phosphorylate lipid molecules, both on the plasma membrane as well as on the membranes of the organelles, changing their reactivity and localization. They represent about 10% of the human kinome [6]. The lipid kinases modulate lipid-based second messengers and lipid signaling pathways, playing a crucial role in the regulation of various cellular processes, such as lipid metabolism, signal transduction, and membrane trafficking. Lipid kinases are categorized as sphingosine and ceramide kinases, diacylglycerol kinases, phosphatidylinositol 3-kinases, lysophosphatidic acid kinases, and choline kinases. In this review, we will give a rapid overview of these kinase subfamilies with a focus on retinal degenerative diseases.

### 3.2.1. The Sphingosine Kinases

The sphingosine kinases have gained more attention in the field of retinal diseases in recent years for their important role in the regulation of the bioactive sphingolipids—sphingosine (Sph) and ceramide (Cer). Sph and Cer are key signaling molecules in the regulation of retinal cell homeostasis. Abnormal Cer promotes inflammation in endothelial and retinal pigment epithelium, causing photoreceptor death. Therefore, deciphering the regulatory mechanisms of these sphingolipids is crucial for the development of a therapeutical approach for retinal degenerative diseases.

The sphingosine kinases 1 and 2 (SPHK1/SPHK2) are the key enzymes in the phosphorylation of sphingolipids in sphingosine 1-phosphate (S1P). SPHK1/SPHK2 and sphingosine 1-phosphate receptors (S1PRs) are expressed in mammalian retinal cells. Deletion of both Sphk1 and Sphk2 in mice causes vascular and neuronal deficits and embryonic mortality, while single deletion of either Sphk1 or Sphk2 does not cause an obvious phenotype due to a compensatory mechanism of the other sphingosine kinase. However, Sphk2 single deletion is sufficient to sensibilize mice to oxygen-induced retinopathy, clearly demonstrating that sphingolipid signaling has an important role in retinal homeostasis. S1P is essential for the survival and cell fate of the photoreceptors and retinal ganglion cells during embryogenesis. In contrast, recent studies have shown that the expression of SPHK1, S1PR2, and S1PR3 increased immediately after light damage and that S1P is detrimental to the retina after an insult, promoting the migration of Müller glial cells and stimulating neovascularization and fibrosis, therefore contributing to the inflammation component present in retinopathies and AMD [20,21].

The ceramide kinase (CerK), which mediates the production of the ceramide 1-phosphate (C1P) from ceramide, is ubiquitously and highly expressed in the retina. Ceramide is a potential second messenger in the TNF signaling pathway, inhibiting cell growth and promoting photoreceptor cell death upon stress conditions. Like for Sph and S1P, C1P controls the opposing processes; i.e., proliferation and cell survival. It is also interesting to mention that deacetylation of ceramide gives sphingosine, while sphingosine and complex sphingolipids can generate ceramide through catabolic pathways [22]. Finally, ceramide promotes inflammation of endothelial and RPE cells in retinopathies and AMD [22].

### 3.2.2. The Diacylglycerol Kinase (DGK)

The retina of mammalian animals expresses several isoforms of the DGK, an enzyme that converts diacylglycerol to phosphatidic acid, thus suppressing the diacylglycerol-mediated intracellular signal transduction. Studies have demonstrated that neuronal cells exposed to hypoxia present an increase in the activity of DGK, inducing higher levels of phosphatidic acid, a positive regulator of mTOR signaling. In contrast, the specific pharmacological inhibition of the activity of the DGK inhibits the transcription mediated by hypoxia-inducible transcription factor (HIF-1$\alpha$), thereby suppressing the activity of the angiogenic factor vascular endothelial growth factor (VEGF). This is a clear example of the promising therapeutic potential of a specific kinase inhibitor, used here to improve neurovascular recovery in a model of retinal ischemia, by downregulating HIF-1$\alpha$ and VEGF [23,24].

### 3.2.3. The Phosphoinositide Kinases (PIKs)

There are 19 phosphoinositide (PI) kinases, subdivided into PI 3-kinases (PI3Ks), PI 4-kinases (PI4Ks), phosphatidylinositol phosphate (PIP) kinases (PIPKs), and phosphatidylinositol phosphate phosphatases, in mammals. The phosphoinositide kinases are highly expressed in the retina, where they have various functions ranging from regulating the chemotaxis of immune cells, such as lymphocytes or macrophages; controlling vesicles and protein trafficking [25]; and modulating signaling pathways involved in cell growth, proliferation, and survival [25–29].

### 3.3. The Protein Kinases

Protein kinases can phosphorylate other protein substrates. The protein kinases are the largest family of enzymes and represent more than 1.5% of the human genome. To date, more than 500 different protein kinases have been identified and/or characterized. As for the previous category of kinases, protein kinases provide the basis for a complex signaling network that transmits and coordinates the response to extracellular stimuli [30].

The eukaryotic protein kinases can be divided into two groups based on their structure: the "eukaryotic" (ePK) and the "atypical" protein kinases (aPKs). The aPKs (alpha kinases, phosphatidyl inositol 3′ kinase-related kinases (PIKKs), pyruvate dehydrogenase kinases (PDHKs), right open reading frame (RIO) kinases) do not share clear sequence similarity with ePKs. The ePKs can be classified into eight families of protein kinases based on their known modes of regulation, their similarities between catalytic domains, and the presence of accessory domains: tyrosine kinase (TK); tyrosine kinase-like (TKL) kinases; STE7-, STE11-, and STE20-related (STE) kinases; casein kinase 1 (CK1); protein kinase A-, protein kinase G-, and protein kinase C-related (AGC) kinases; $Ca^{2+}$/calmodulin-dependent kinases (CAMKs); cyclin-dependent kinases (CDKs), mitogen-activated protein (MAP) kinases, glycogen synthase kinases (GSKs), and CDK-like kinases (CMGC kinases); and receptor guanylyl cyclase (RGC) [31].

#### 3.3.1. The Tyrosine Kinases (TKs)

The tyrosine kinases phosphorylate proteins on tyrosine residues. TKs can be separated into the receptor TKs (RTKs, transmembrane), such as EGFR or VEGFR, and the non-receptor TKs (cytosolic TKs), such as Src, Abl, or JAK kinases. Upon extracellular ligand binding, the RTKs form oligomers and activate their intracellular kinase domains, leading to the transmission of a signal into the cytoplasm. The TKs represent most targets for kinase inhibitors currently used clinically [32–34].

Multitarget tyrosine kinase inhibitors have been developed to block the downstream activation of many RTKs in oncology. Some of these multitarget inhibitors targeting the VEGF receptors are already approved as first-line treatment options to curb the progression and invasiveness of aggressive cancers, such as the metastatic renal cell carcinoma. Blocking the abnormal activation of RTKs in the retina is now being investigated as an option to limit the proliferation and invasiveness of abnormal blood vessels in the retinal tissue mediated by the RTKs VEGF, platelet-derived growth factor (PDGF), fibroblast growth factor (FGF), and angiopoietin in multiple retinal vascular diseases, such as aged-macular degeneration or diabetic retinopathy [35]. Several clinical studies are exploring different inhibitors targeting VEGF, PDGF, FGF, and the angiopoietin receptor Tie, such as vorolanib, axitinib, and sunitib malate, as well as alternative delivery methods (for example, oral delivery of vorolanib, bioerodible implants loaded with vorolanib, and a bioresorbable hydrogel intravitreal implant delivering micronized axitinib) to reduce the burden of frequent intravitreal injections of anti-VEGF agents and to improve the real-world outcomes in the treatment of aged-macular degeneration [36,37]. Multitarget inhibitors of tyrosine kinases are also being tested in the context of diabetic retinopathy or oxygen-induced retinopathy. Sergeys et al. have conducted activity-based tyrosine kinome profiling on retinal tissue from a mouse model of early and advanced diabetic retinopathy (Akimba mouse model), unveiling the central role of the Src-FAK kinases. The toxicity and anti-angiogenic effects of several Src and FAK kinase inhibitors with different specificity profiles were tested with in vitro and ex vivo models [38]. It would therefore be interesting to continue to study the regulatory mechanism of these kinases in the retina to develop new and integrative therapeutic approaches for patients with ocular diseases.

#### 3.3.2. The Tyrosine Kinase-like (TKL) Kinases

The TKL kinases represent the majority of serine/threonine kinases. This family contains both nonreceptor and receptor kinases. It is interesting to report that a genome-wide RNA interference-based forward genetic screen, realized with small interfering RNA tar-

geting all human kinases, has identified clusters of TKL kinases essential for the regulation of axodendritic and synaptic integrity and synaptic connectivity. Almost 50% of the 59 positive regulators of neurite outgrowth identified were TK/TKL receptors, while none of the 66 negative regulators of neurite outgrowth belonged to the TK or TKL families. Similarly, none of the 79 kinases involved in the inhibition of the lysophosphatidic acid-induced neurite retraction were TKs/TKL kinases [39].

### 3.3.3. The Sterile (STE) Protein Kinases Group

The STE group includes homologs of the yeast kinase sterile 7 protein (Ste7 or MAP2K), sterile 11 protein (MAP3K), and sterile 20 protein (Ste20 or MAP4K), which render yeast sterile if deleted. Several studies have demonstrated that Ste kinases play an important role in retinal development and homeostasis. Among the Ste-20 kinases, the mitogen-activated protein kinase kinase kinase kinase-4 (MAP4K4), identified as one of the pivotal kinases in *Saccharomyces cerevisiae* and well known for its functions in the regulation of cell proliferation, inflammation, and metabolism, has been also associated with neurodegeneration and neural remodeling. Map4k4 deletion causes early embryonic lethality. Moreover, MAP4K4 plays a critical role during neurogenesis and retinal photoreceptor differentiation via the regulation of the transcription factor c-JUN. MAP4K4 misregulation has been linked to neonatal retinal angiogenesis and oxygen-induced retinopathy [40]. Zhang et al. recently demonstrated that, after NMDA-induced damage of the retina, mammalian MAP4K4, MAP4K6, and MAP4K7 repress the initiation of cell cycle reentry, as well as Müller glia reprogramming into photoreceptor progenitor cells, mediated by the co-transcription factor YAP, thus impeding the regenerative capacities of the mammalian retina (Figure 4) [41]. In non-mammalian vertebrate models, like zebrafish, Müller glial cells present the capability to dedifferentiate in self-renewing retinal progenitors, which migrate to the lesion site and differentiate into the new photoreceptors [42]. To mimic the regenerative capability of the zebrafish co-transcription factor YAP, it has been proposed to inhibit the ste20 kinases MAP4K4/6/7 using a small molecule inhibitor. Mouse Müller glial cells treated with a small molecule inhibitor of MAP4K4/6/7 present the ability to initiate cell cycle reentry and produce a retinal progenitor-like cell state after retinal damage. The mouse Müller glial cells present spontaneous trans-differentiation into retinal neurons expressing amacrine and retinal ganglion cell markers after inhibitor withdrawal (Figure 4, left). This study clearly demonstrates the importance of understanding the functions of kinases in retinal degenerative diseases and the therapeutical potential of their manipulation, here via a kinase inhibitor that makes it possible to access the capacity of retinal Müller glia to reprogram into retinal progenitors and retinal regeneration in adult mammals [41].

### 3.3.4. The Casein Kinase 1 (CK1) Family

The CK1s are serine/threonine-selective enzymes that are constitutively expressed. They are mostly involved in cytoskeletal function and transcriptional regulation. Recently, it has been shown that the CKI inhibitor D4476 regulates both autophagy and apoptosis in immortalized RPE cells [43].

### 3.3.5. The AGC Kinases

The AGC group is named for the enzyme families PKA, PKC, and PKG [44]. The human AGC protein kinase group contains more than 60 protein kinases, classified into 14 subfamilies: PDK1, AKT/PKB, SGK, PKA, PKG, PKC, PKN/PRK, RSK, NDR, MAST, YANK, DMPK, GRK, and SGK. Amongst them, the AKT/PKB kinases are particularly important for protein synthesis, glucose metabolism, and the survival of retinal cells under stress conditions. AKT/PKB kinases are thus potential interesting targets for the treatment of retinal diseases, such as chronic ocular hypertension, diabetic retinopathy, photoreceptor light damage, or neovascularization [45]. It has also been demonstrated that the AGC kinases ROCK1 and PKN1 are important players in the retinal degeneration observed in retinitis pigmentosa [39].

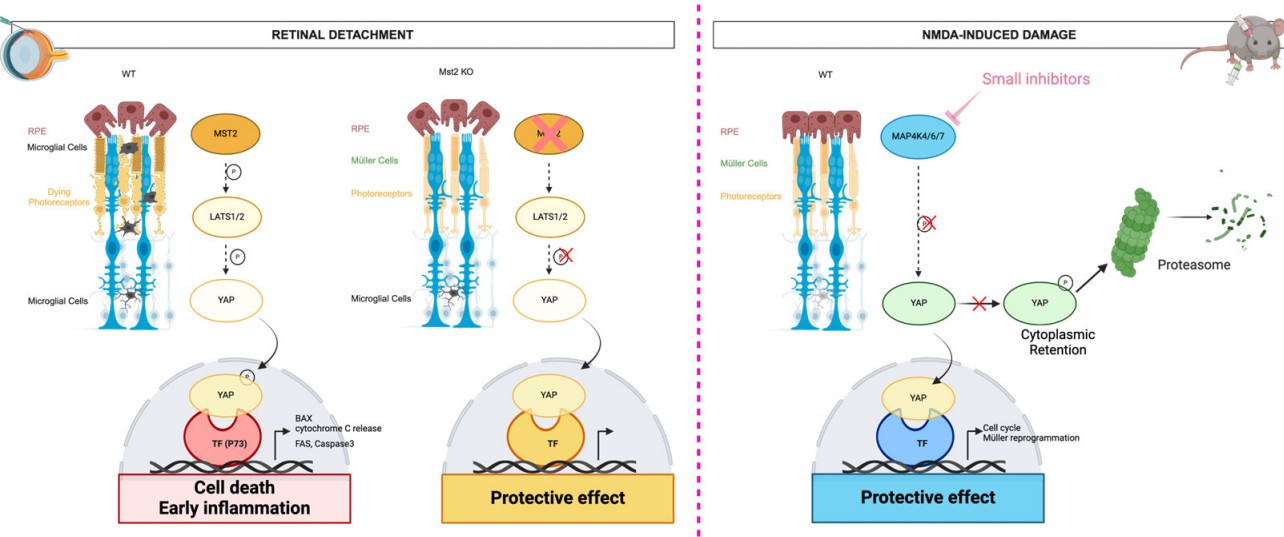

**Figure 4.** Schematic of the regulation of YAP co-transcription factor by (left) MAP4K4/6/7 after NMDA-induced damage of the retina and (right) MST2 after retinal detachment. The mammalian MAP4K4/6/7 induces the phosphorylation of YAP (P-YAP), leading to its cytoplasmic retention and degradation by the proteasome and, consequently, the death of the photoreceptors. When the mammalian MAP4K4/6/7 are inactivated by the injection of small inhibitors, YAP is not phosphorylated and can be translocated into the nucleus. The nucleus form of YAP can activate Müller cells' reprogramming into retinal progenitors to repopulate the photoreceptors. In a similar manner, MST2 leads to the phosphorylation of YAP (P-YAP) and translocation of P-YAP to the nucleus after retinal detachment. Nucleus P-YAP can activate an early inflammatory response, as well as cell death mechanisms. Mst2 knock-out mice present neither translocation of P-YAP to the nucleus nor photoreceptor cell death after retinal detachment. Created with BioRender.com.

### 3.3.6. The Ca$^{2+}$/Calmodulin-Dependent Protein Kinases (CAMKs)

The CAMKs are a group of serine/threonine-specific protein kinases activated by the increase in the intracellular concentration of calcium ions. The cell cycle checkpoint kinases, CHK1 and CHK2, belong to the CAMKs. Upon DNA damage, they initiate a phosphorylation cascade that leads to cell cycle arrest and DNA repair [46,47].

### 3.3.7. The CMGC Group

The CDK/MAPK/GSK3/CLK (CMGC) kinase group contains cyclin-dependent kinases (CDKs), mitogen-activated protein kinases (MAPKs), glycogen synthase kinases (GSKs), and CDC-like kinases (CLKs), which are key players in numerous cellular signaling pathways regulating cell cycle regulation, proliferation, differentiation, apoptosis, gene expression regulation, and inflammation [6]. Two CMGC kinases, male germ cell-associated kinase (MAK) and intestinal cell kinase (ICK), are implicated in the regulation of the intraflagellar transport turnaround at the ciliary tip and in control of the ciliary length in mammalian cells. As the outer segment, the light-sensory structure of the retinal photoreceptor cells is formed from the primary cilia in photoreceptor precursors, and ciliary dysfunction plays an important role in retinal degeneration and blindness [48].

### 3.3.8. The Receptor Guanylyl Cyclase (RGC) Group

This group of kinases is the smallest one and is indeed constituted by pseudokinases. These enzymes are missing the critical residues for phosphate transfer and are only able to convert GTP to cyclic GMP [6].

### 3.4. The Serine/Threonine (Ser/Thr) Kinases

The Ser/Thr kinases represent one of the largest groups of kinases, with more than 300 members encoded in the human genome. They are responsible for 90,000 sites of serine and threonine phosphorylation and linked to numerous biological processes and diseases. Ser/Thr kinases phosphorylate the amino acids serine and/or threonine in their substrates, as serine and threonine amino acids share a similar side chain. The specificity of the Ser/Thr kinases is determined by their consensus sequence, an essential and specific sequence of amino acids that comprise the phosphor-acceptor site and the flanking residues. That consensus sequence contacts several key amino acids located in the catalytic domain of the kinase, often via hydrophobic forces and ionic bonds. The interaction between the consensus sequence of the substrate and the catalytic domain of a kinase is usually not specific to a single substrate but is broad enough to include a whole "substrate family" sharing common recognition sequences. Moreover, most of the catalytic domain of the Ser/Thr kinases is highly conserved across living organisms, giving us easy access to it by using eucaryotic unicellular models [49,50]. The serine/threonine kinases are regulated by various chemical signals, such as lipids or cyclic AMP, and by cellular events, such as oxidative stress, mechanical stress (extracellular matrix stress, pressure on the cell membrane, cell density), and cell polarity. They play an important role in neurodegenerative diseases, which highlights the importance of studying their functions and regulation [51–53].

The Nuclear Dbf2-Related Kinases

Amongst the serine/threonine kinases, the nuclear Dbf2-related kinases are ancient, widespread, and highly conserved kinases amongst living organisms and a subclass of the AGC protein kinases [52]. First studied in budding yeast, the nuclear Dbf2-related kinases are key players in the cell cycle, proliferation and cell death regulation, and asymmetric cell division between mother and daughter cells, thus implying strict control of the cytoskeleton and cell polarity. In drosophila, these kinases are important for proliferation and cell death regulation, neuronal cell fate, and dendritic tiling, branching, and maintenance, which is controlled by cell polarity and is similarly observed in mammals [53]. The nuclear Dbf2-related kinases are also implicated in controlling cell migration, trafficking, endocytosis, and autophagy in numerous tissues, including the central nervous system and the ocular system [53]. The nuclear Dbf2-related kinases are activated by phosphorylation of their activation region, located in the C-terminal of the kinase catalytic domain [54]. The nuclear Dbf2-related kinase family possesses four core kinases: large tumor suppressor (LATS) 1 and LATS2 and nuclear Dbf2-related (NDR)1 and NDR2, also known as serine-threonine kinase 38 (STK38) and serine-threonine kinase 38 like (STK38L), respectively. LATS1 and LATS2 are considered the canonical branch of the Hippo pathway, while NDR1 and NDR2 are understudied and considered the noncanonical branch of the Hippo pathway. LATS1 and LATS2 are the paralogous terminal kinases of the well-studied *Drosophila melanogaster* signaling, the Hippo pathway (also called the Salvador–Warts–Hippo pathway) [53]. The core of the canonical Hippo pathway is a kinase cascade, starting with the highly conserved Ste20-like kinases MST1 and MST2 (ortholog of Drosophila Hippo) and ending with the LATS1 and LATS2 kinases, which sequentially phosphorylate their substrates. In mammals, the main substrates of the Hippo signaling are the paralogous transcription co-factors Yes-associated protein (YAP) and WW domain-containing transcription regulator 1 (TAZ), which are cytoplasmic. They are translocated in the nucleus to promote transcription when dephosphorylated. The noncanonical Hippo kinase cascade is composed of the upstream Ste20-like kinases MST1, MST2, and MST3, implicated in the activation of the core kinases NDR1 and NDR2 and ending with the phosphorylation of their substrates (Figures 4 and 5B).

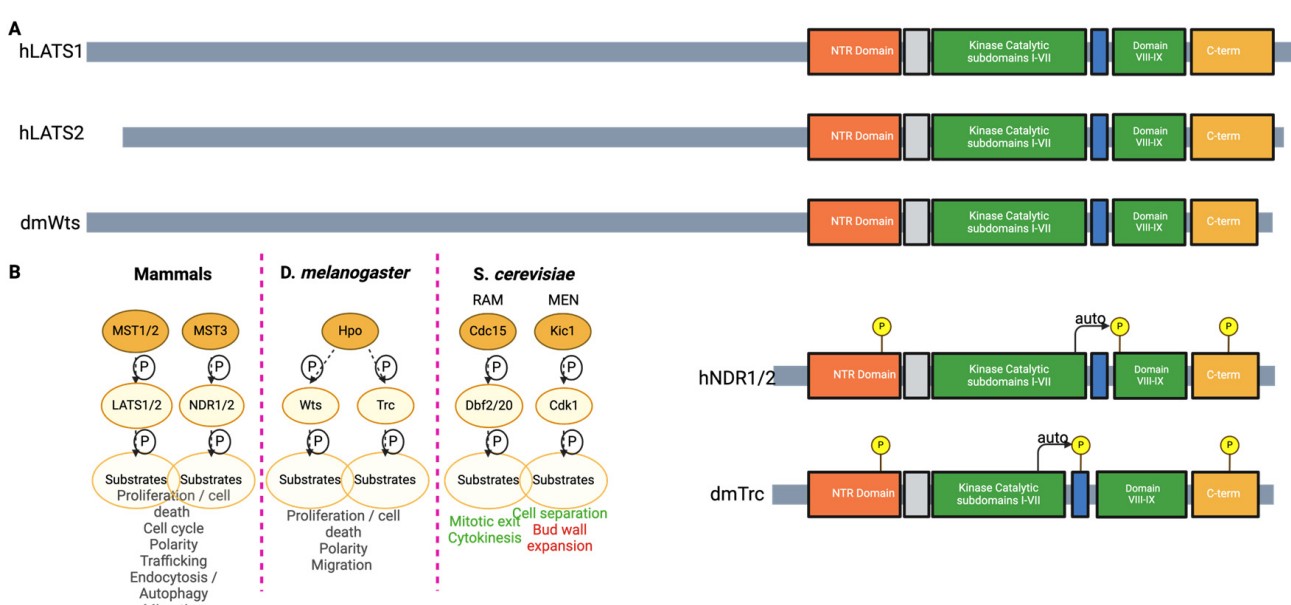

**Figure 5.** The nuclear Dbf2-related kinases amongst living organisms. (**A**) Schematic of the structure of the human and fly nuclear Dbf2-related kinases LATS1, LATS2, NDR1, and NDR2. (**B**) Summary of mammalian NDR/LATS kinase signaling pathway in mammals, flies (*D. melanogaster*), and budding yeast (*C. cerevisiae*). Created with BioRender.com.

The mammalian STE20-like kinases MST1 and MST2 are regulators of cell fate, cell death, and proliferation. They share 98% identity in their catalytic domain and 60% identity in their regulatory region, which indicates specificity in their expression and triggers [55]. Overall, MST2 is closer to the fly protein Hippo than MST1 (Figure 5A). In the last decade, it has been demonstrated that MST2, but not MST1, plays a crucial role in the photoreceptor cell death after retinal detachment. Indeed, after retinal detachment, mice with Mst2 deleted present less phosphorylated co-transcription YAP in the nucleus compared with WT mice, causing significantly less photoreceptor cell death. Mice with Mst2 deleted also demonstrate significantly lower proinflammatory cytokines expression after retinal detachment (Figure 4, right). Therefore, MST2, but not MST1, could be a future protective target [55,56].

## 4. Discussion

Reversible modifications of proteins, such as the addition of a γ-phosphate from ATP by kinases, also known as phosphorylation, are key to cellular adaptability to the environment. Conversely, perturbations of such adaptability are the cause of many diseases. Kinases are one of the largest groups of enzymes in living organisms and therefore represent interesting targets for therapeutical approaches. Currently, about one third of all protein targets under research in the pharmaceutical industry are kinase-based, demonstrating the supreme importance of understanding the structure, functions, and regulation of kinases with regard to disease treatment.

Kinase-targeted therapies against malignancies such as breast and lung cancer started in the early 1980s and have been extensively developed since that time. Today, there are over 40 kinase inhibitors already approved by the FDA, and more than 150 kinase-targeted drugs in different phases of clinical trials [34,57,58]. Kinase-targeted therapies in cancer treatment permit more accurate targeting of the tumor cells by playing on the differences between the specific tumor genetics and tumor microenvironment compared to the genetics and microenvironment of healthy cells. However, the field of neurodegenerative disorders, and especially ophthalmology, is far behind in the use of kinase-targeted therapies [35,37,45,51,59].

In recent years, with information and inspiration from the research on cancer, more and more kinase-targeted therapies targeting tyrosine kinases, STE protein kinases, or serine/threonine kinases have been adapted to the neuronal environment to induce regenerative capabilities similar to the ones observed in non-mammalian vertebrae, control neural inflammation, or curb the abnormal angiogenesis observed in several ocular diseases, such as age-related macular degeneration or diabetic retinopathy. For example, in the last decade, nilotinib, an FDA-approved inhibitor of the kinase cellular abelson murine leukemia viral oncogene homolog 1 (c-Abl, also named ABL1), has been studied as a potential treatment for Parkinson's and Alzheimer's diseases [60]. Indeed, earlier studies demonstrated the positive effect of c-Abl on neuronal development and the association between its aberrant activation and Parkinson's and Alzheimer's diseases [55]. It is interesting to mentioned that the Hippo signaling pathway—an essential player in the regulation of numerous cellular processes, such as cell proliferation, tissue regeneration, and immune response, centered around several serine/threonine kinases—could be of interest in exploring new therapeutic avenues for the prevention or management of ophthalmic diseases by targeting these kinases [56].

However, despite the spectacular results shown by kinase-targeted therapies in cancer treatment [61], their application to retinal degenerative diseases such as AMD, diabetic retinopathy, or retinitis pigmentosa has proven difficult. It is important to point out that retinal degenerative diseases encompass a very diverse group of disorders with different underlying genetic mutations and pathophysiological mechanisms. Kinase inhibitors may only be effective in subsets of patients with specific molecular targets, limiting their broad applicability. Therefore, kinase-targeted therapies for retinal diseases are confronted with several limitations associated with the specificity of the kinase inhibitors, which often target multiple kinases, thereby generating off-target effects and severe adverse effects on normal functions of the neuronal cells. Moreover, retinal treatments often require challenging administration routes to circumvent the blood–retinal barrier. In particular, intraocular injections are complex procedures that can engender secondary retinal issues (pain/foreign body sensation, possibly due to dry eye or corneal abrasion, infection, subconjunctival or vitreous hemorrhage, retinal tear or retinal detachment, or cataracts) [62].

As for cancer therapy, kinase-targeted therapies for retinal diseases may initially show promise in preclinical studies or early-phase clinical trials while presenting no or modest efficacy in halting disease progression or reversing vision loss in the real world. This difference in visual acuity outcomes between those observed in clinical trials and in the real world is even more pronounced in therapies for retinal diseases based on intravitreal injections due to undertreatment outside of the clinical trial environment. Thus, a lapse in treatment of neovascular AMD of more than 3 months has been shown to lead to irreversible vision loss for a minimum of 1 year [36].

Addressing these limitations requires interdisciplinary collaboration among researchers, clinicians, and pharmaceutical companies to develop more selective kinase inhibitors, refine drug delivery strategies, and identify patient subpopulations most likely to benefit from this therapeutic approach. Additionally, exploring combination therapies targeting multiple pathways implicated in retinal degeneration may enhance treatment efficacy and minimize the risk of resistance. Finally, developing effective delivery methods to the retina is crucial for the treatment of retinal degenerative diseases. Various approaches are currently being explored. Hydrogels composed of crosslinked polymeric networks can absorb and retain large amounts of liquid and offer a promising platform for sustained drug delivery to the retina. Hydrogels can be engineered to release therapeutics in a controlled manner, providing prolonged drug exposure while minimizing systemic side effects [63]. Drug-loaded contact lenses releasing drugs directly onto the ocular surface are being investigated as a non-invasive and patient-friendly approach for ocular drug delivery. Contact lenses offer the advantage of easy application and removal, making them suitable for both acute and chronic treatments [64,65]. Sustained-release implants can be designed to release drugs continuously over an extended period and surgically placed in the vitreous

or subretinal space. They permit reduced treatment frequency compared to traditional eye drops or injections [66]. Gene therapy holds promise for treating inherited retinal diseases by delivering therapeutic genes directly to the retina. Viral vectors, such as adeno-associated viruses (AAVs), can be engineered to carry corrective genes and safely deliver them to target cells in the retina. Gene therapy offers the potential for long-term treatment benefits by addressing the underlying genetic cause of retinal diseases [64]. Topical eye drops and systemic medications can be used for treating certain retinal conditions, such as diabetic retinopathy or retinal vein occlusion. While topical therapies offer the advantage of non-invasiveness and ease of administration, they may have limited penetration into the posterior segment of the eye [67]. Systemic medications, on the other hand, can reach the retina via systemic circulation but may cause systemic side effects. Continued research and innovation in drug delivery technologies are essential to advance the field of retinal therapeutics and improve patient care.

## 5. Conclusions

Although drawing parallels between cancer and neurodegenerative disorders is complicated since neuronal tissues are often composed of quiescent cells while cancer cells present abnormal over-proliferation, there are some common mechanisms in both disorders. Interestingly, the Hippo signaling pathway can link these two important disorders. It is thus important to improve our current understanding of the specific regulation of the Hippo pathway in neural tissues and (neuro)degenerative diseases. In a broader sense, it is crucial to better understand the kinome in healthy and diseased neural systems, as has been achieved over more than 50 years in the field of cancer, to develop effective and reliable therapeutic targets to prevent or delay the onset of retinal degenerative diseases.

**Author Contributions:** Conceptualization: H.L. and P.F.S.; Writing—review and editing: H.L., A.F.A., and P.F.S.; Figures: H.L. and P.F.S.; Funding acquisition: H.L., P.F.S. and A.F.A. All authors have read and agreed to the published version of the manuscript.

**Funding:** This manuscript was supported by funds from the Fundação para a Ciência e a Tecnologia, Portugal (individual grant 2022.06170.PTDC; institutional grants UIDB/04539/2020, and UIDP/04539/2020).

**Institutional Review Board Statement:** Not applicable.

**Informed Consent Statement:** Not applicable.

**Data Availability Statement:** Not applicable.

**Acknowledgments:** We are indebted to colleagues from the University of Coimbra who provided insightful advice and language help and proofread this article.

**Conflicts of Interest:** The authors declare no conflicts of interest.

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
