# Peer review of "The Importance of Kinases in Retinal Degenerative Diseases"

_2813-3757, doi:10.3390/kinasesphosphatases2010006_

Round 1

Reviewer 1 Report

Comments and Suggestions for Authors

The review highlights several important issues on kinases and vision. The manuscript is well written and covers different areas.

It would be interesting to see more comments on macular degeneration, and kinases see doi: 10.1080/14728214.2023.2259790 since the incidence of this disease has increased in the last years, and it has been related to several vascular and neurovascular conditions. 

The article is an interesting. It is related to degenerative diseases and gives basic information on different mechanisms involved in retinal degeneration.

The article provides a general view of the signals involved in normal retinal physiological responses and the conditions that appear when the signal pathways are impaired. The information is important mainly for clinicians to understand the mechanism of disease, and that is the strength of the article. The article, however, does not provide a response to therapeutics, and it is understandable since the scope of the journal is not pharmacology, and the article already has important information.

The new elements of the review are the updated information and the figures which are important for the readers to properly understand the subject. Finally, the references are appropriate and as I commented the figures and tables are informative. In my opinion, the short list of comments is because the review, in general, is acceptable for publication.

Comments on the Quality of English Language

Minor grammatical mistakes

Author Response

Dear reviewer,

we would like to thank you for your time and your constructive comments, which we gave full consideration.

First, we would like to thank you for pointing out to us the review paper on macular degeneration, and kinases (“Das N, Chaurasia S, Singh RP. A review of emerging tyrosine kinase inhibitors as durable treatment of neovascular age-related macular degeneration. Expert Opin Emerg Drugs. 2023 Dec;28(3):203-211. doi: 10.1080/14728214.2023.2259790. Epub 2023 Oct 12. PMID: 37796039.”). We have integrated that comment in our manuscript (Lines 332 to 341, in red)

In addition, we have also corrected some typos in the manuscript, and some other minor adjustments. We hope that these corrections have adequately address your concerns.

Sincerely,
Hélène Léger, PhD
Retinal Dysfunction & Neuroinflammation Lab
Coimbra Institute for Clinical and Biomedical Research (iCBR), Faculty of Medicine University of Coimbra

Reviewer 2 Report

Comments and Suggestions for Authors

The subject discussed in the article titled "The Significance of Kinases in Retinal Degenerative Diseases" authored by Santos et al. appears to be quite intriguing. Overall, this paper is crafted with conciseness and organization. It is easy to read, well-structured, and presented, making it potentially appealing to both readers and researchers alike. However, I would like to offer the following suggestions for enhancements that could be implemented:

1.    Kindly provide alternative titles for the initial subsections.

2.    Could you please confirm the originality of figures 2, 4, and 5? If not original, do we possess the necessary permissions for their usage?

3.    Numerous paragraphs in the manuscript lack proper references. For instance, sections such as lines 164-176 and 248-254 remain unreferenced. A thorough revision of the entire article is warranted in this regard.

4.    I would like the limitations and future research directions to be addressed before the conclusions section.

5.    The References section contains various bibliographic entries with incomplete information, notably lacking page numbers or authors' names. I recommend filling in these missing details for completeness.

Comments on the Quality of English Language

Minor editing of English language required.

Author Response

Dear reviewer,

we would like to thank you for your time and your constructive comments, which we gave full consideration. Below we present our responses to each comment, as well as the actions taken to address them. 

We have provided alternative titles for the initial subsections, and we hope that we have modified the appropriate subsections.

The figure 2 was based on two figures from the following article: Das S, Chen Y, Yan J, Christensen G, Belhadj S, Tolone A, Paquet-Durand F. The role of cGMP-signalling and calcium-signalling in photoreceptor cell death: perspectives for therapy development. Pflugers Arch. 2021 Sep;473(9):1411-1421. doi: 10.1007/s00424-021-02556-9. Epub 2021 Apr 16. PMID: 33864120; PMCID: PMC8370896 which should have been cited but which citation was missing. We thank the reviewer #2 for point that issue to us. To remedy to that situation, we added the statement that the figure 2 is adapted from the aforementioned article. Moreover, we improved figure 2 with the proper citations (Li, S.; Ma, H.; Yang, F.; Ding, X. cGMP Signaling in Photoreceptor Degeneration. Int. J. Mol. Sci. 2023, 24, 11200. https://doi.org/10.3390/ijms241311200; Kim JJ, Lorenz R, Arold ST, Reger AS, Sankaran B, Casteel DE, Herberg FW, Kim C. Crystal Structure of PKG I:cGMP Complex Reveals a cGMP-Mediated Dimeric Interface that Facilitates cGMP-Induced Activation. Structure. 2016 May 3;24(5):710-720. doi: 10.1016/j.str.2016.03.009. Epub 2016 Apr 7. PMID: 27066748; PMCID: PMC4856591).

Figures 4 and 5 are original.

We have revised the entire article for proper references. All new additions are in red. We would like to thank you for your comment.

We have discussed (part 4. Discussion) the limitations of kinase-targeted therapies and how these limitations are currently addressed or can be improved by future research.

We have corrected the References section and the English language. In addition, we have also corrected some typos in the manuscript, and some other minor adjustments. We hope that these corrections have adequately address your concerns.

Sincerely,
Hélène Léger, PhD
Retinal Dysfunction & Neuroinflammation Lab
Coimbra Institute for Clinical and Biomedical Research (iCBR), Faculty of Medicine University of Coimbra